# Cancer disrupts sex hormone-inflammation relationships: Analysis of ALI in males from NHANES 2007–2018

Wenyao Xie[1], Zhenjun Zhang[1], Dan Zhao[2], Ya Zhang[1], Caiting Feng[3], Yicong Zhou[4]*

1 Department of Oncology, Handan Central Hospital, Hebei, Handan, P.R. China, 2 Department of Emergency, Handan First Hospital, Hebei, Handan, P.R. China, 3 Department of Thoracic Surgery, Handan First Hospital, Hebei, Handan, P.R. China, 4 Department of Neurology, Handan Central Hospital, Hebei, Handan, P.R. China

* zyc890915@163.com

## Abstract

### Background

To investigate the correlation between sex hormone levels and Advanced Lung Cancer Inflammation Index (ALI) in male cancer populations, and compare with non-cancer male populations, in order to elucidate the potential regulatory role of sex hormones in cancer-related inflammation.

### Methods

We analyzed data from NHANES 2007–2018, including 11,280 males (1,135 cancer patients, 10,145 non-cancer controls). Serum testosterone and estradiol were measured using ID-LC-MS/MS. ALI was calculated as BMI(kg/m²) × serum albumin(g/dL)/ NLR. We employed weighted multivariate regression models with progressive adjustment for confounders and conducted stratified analyses by cancer status, age, and race/ethnicity. Restricted cubic splines explored non-linear relationships.

### Results

Cancer patients had higher mean age (68.70±11.85 vs. 48.36±17.00 years), lower testosterone levels (377.35±204.26 vs. 415.02±183.92 ng/dL, P<0.001), and lower ALI values (64.86±118.61 vs. 72.68±185.25, P=0.053) than non-cancer controls. In fully adjusted models, each 10 ng/dL increase in testosterone was associated with a 3.0% decrease in ALI (OR=0.970, 95%CI: 0.962–0.978, P<0.001), while each 1 pg/mL increase in estradiol was associated with a 60.3% increase in ALI (OR=1.603, 95%CI: 1.318–1.949, P<0.001). Notably, these associations were significant only in non-cancer populations (testosterone: OR=0.97, P<0.001; estradiol: OR=1.64, P<0.001), and generally absent in cancer patients except for older cancer patients (≥60 years) where testosterone maintained a significant negative correlation with

**Data availability statement:** AIThe data underlying this study are publicly available from the National Health and Nutrition Examination Survey (NHANES) database maintained by the Centers for Disease Control and Prevention (CDC). Data from NHANES 2007-2018 can be accessed directly from the CDC website at https://www.cdc.gov/nchs/nhanes/?CDC_AAref_Val=https://www.cdc.gov/nchs/nhanes/index.htm. The specific datasets used in this study can be downloaded from https://wwwn.cdc.gov/nchs/nhanes/continuousnhanes/default.aspxbyselectingtherelevantsurveycycles(2007-2008,2009-2010,2011-2012,2013-2014,2015-2016,and2017-2018). These datasets are available without restriction for research purposes. No additional data beyond what is available in these public repositories was used in this study.

**Funding:** The author(s) received no specific funding for this work.

**Competing interests:** The authors declare that they have no competing interests.

ALI (OR=0.96, P = 0.020). The testosterone-ALI negative correlation was strongest in younger individuals (20–39 years), while the estradiol-ALI positive correlation weakened with age. Estradiol exhibited significant non-linear relationship with ALI (P = 0.027), with multiple inflection points suggesting concentration-dependent effects.

## Conclusion

This study systematically reveals for the first time the association between male sex hormone levels and ALI and the moderating effect of cancer status. The negative correlation between testosterone and ALI and the positive correlation between estradiol and ALI are significant in non-cancer populations but generally absent in cancer patients, suggesting that cancer may profoundly alter the relationship between sex hormones and inflammation. These findings suggest cancer may fundamentally alter sex hormone-inflammation relationships, providing new insights into hormone-mediated inflammatory regulation in health and disease.

## Introduction

Systemic inflammatory response plays a critical role in the initiation, progression, and prognosis of cancer [1]. Chronic inflammation facilitates the formation of the tumor microenvironment, enhances tumor cell proliferation, invasion, and metastatic potential, and suppresses anti-tumor immune responses [2,3]. Numerous studies have demonstrated that inflammatory markers, such as C-reactive protein (CRP), neutrophil-to-lymphocyte ratio (NLR), and platelet-to-lymphocyte ratio (PLR), are associated with adverse prognosis in various cancer types [4–7]. The Advanced Lung Cancer Inflammation Index [8] is a comprehensive biomarker that integrates albumin, neutrophil count, and lymphocyte count. It was initially proposed by Jafri et al. [9] to evaluate the prognosis of advanced lung cancer patients. ALI has not only demonstrated excellent predictive value in lung cancer, but in recent years, its prognostic assessment significance has been confirmed across multiple malignant tumors [10–12], emerging as a crucial indicator that reflects the tumor-associated systemic inflammatory state and prognostic status.

Sex hormones, particularly androgens and estrogens, possess significant immunomodulatory functions. Androgens typically exhibit anti-inflammatory characteristics, while estrogens demonstrate either pro-inflammatory or anti-inflammatory effects depending on specific physiological conditions [13]. In males, testosterone regulates inflammatory responses by suppressing the production of pro-inflammatory cytokines (such as IL-1β, IL-6, and TNF-α) and promoting the release of anti-inflammatory cytokines (like IL-10) [14]. Moreover, sex hormones can influence the development, differentiation, and function of immune cells, thereby modulating systemic inflammatory states [15]. Numerous researches have demonstrated the intimate association between sex hormones and the occurrence, progression, and prognosis of various cancers. In prostate cancer, activation of the androgen signaling pathway is a critical driving factor of disease progression [16]; in breast cancer, estrogen promotes tumor

growth [17]. However, research on the role of sex hormones in non-gonadal-dependent cancers remains relatively limited. Some studies have suggested that decreased testosterone levels may be associated with an increased risk of certain inflammation-related diseases [18,19], but the relationship between sex hormone levels and inflammatory markers in cancer patients has not been comprehensively investigated.

Currently, systematic research on the relationship between sex hormone levels and ALI remains lacking, especially in comparative studies between cancer and non-cancer populations. Given the important role of inflammation in cancer development and the potential influence of sex hormones in regulating inflammatory processes, exploring the correlation between sex hormones and ALI not only helps to deepen our understanding of cancer pathogenesis but may also provide new perspectives for cancer risk assessment, prognosis determination, and treatment strategies. Therefore, based on the large NHANES database, this study aims to investigate the correlation between sex hormone levels and ALI in male cancer populations, and whether this correlation differs compared to non-cancer male populations. By elucidating these relationships, empirical evidence will be provided by this research for understanding the role of sex hormones in cancer-related inflammation, and new ideas may be offered for clinical practice.

## Materials and methods

### Data source

This study is based on public data from the National Health and Nutrition Examination Survey (NHANES) from 2007–2018 (https://www.cdc.gov/nchs/nhanes/). NHANES is conducted by the National Center for Health Statistics (NCHS) of the Centers for Disease Control and Prevention (CDC) as a nationwide, continuous survey that employs a complex, multi-stage, stratified probability sampling design. It aims to assess the health and nutritional status of non-militarized, non-institutionalized residents in the United States. Data are collected by this survey in two-year cycles through detailed questionnaires, physical examinations, and laboratory tests.

The screening process for research subjects is shown in Fig 1, with the initial NHANES 2007–2018 database containing 59,842 participants. First, participants under 20 years old were excluded, leaving 34,770 individuals. Subsequently, participants with missing demographic information were removed, reducing the sample to 31,206. Ensuring all participants had clear cancer classification information, the sample size became 31,185. Next, participants with missing related indicators (such as albumin, neutrophil count, lymphocyte count, etc.) were systematically eliminated, and we retained 23,613 participants. Since the research focused on the relationship between male sex hormones and ALI, only male participants were included, ultimately resulting in a final sample of 11,280 individuals. This group comprised 1,135 cancer patients and 10,145 non-cancer controls. All included participants were categorized into cancer and non-cancer control groups based on their cancer status. The research protocol was conducted in accordance with NHANES data use policies. Because the study utilized publicly available, de-identified data, additional ethical approval was not required.

Cancer status in our study was determined using the NHANES Medical Conditions Questionnaire (MCQ), which collected self-reported data on medical conditions through personal interviews conducted by trained NHANES staff. Specifically, participants were asked "Has a doctor or other health professional ever told you that you had cancer or a malignancy of any kind?" (variable MCQ220). Those responding "Yes" were classified as having a cancer history, while those responding "No" formed our non-cancer group. Participants with "Don't know" or refused responses were excluded from our analysis. For participants confirming a cancer history, additional information was collected regarding cancer type and age at diagnosis. We excluded individuals with lung cancer to avoid potential confounding with our primary outcome measure.

### Variable definition and measurement

**Measurement methods of sex hormones.**  The measurement of sex hormone levels employs NHANES standard laboratory methods. Serum testosterone and estradiol concentrations were determined through isotope dilution-liquid chromatography-tandem mass spectrometry (ID-LC-MS/MS), a method which the CDC Environmental Health Laboratory

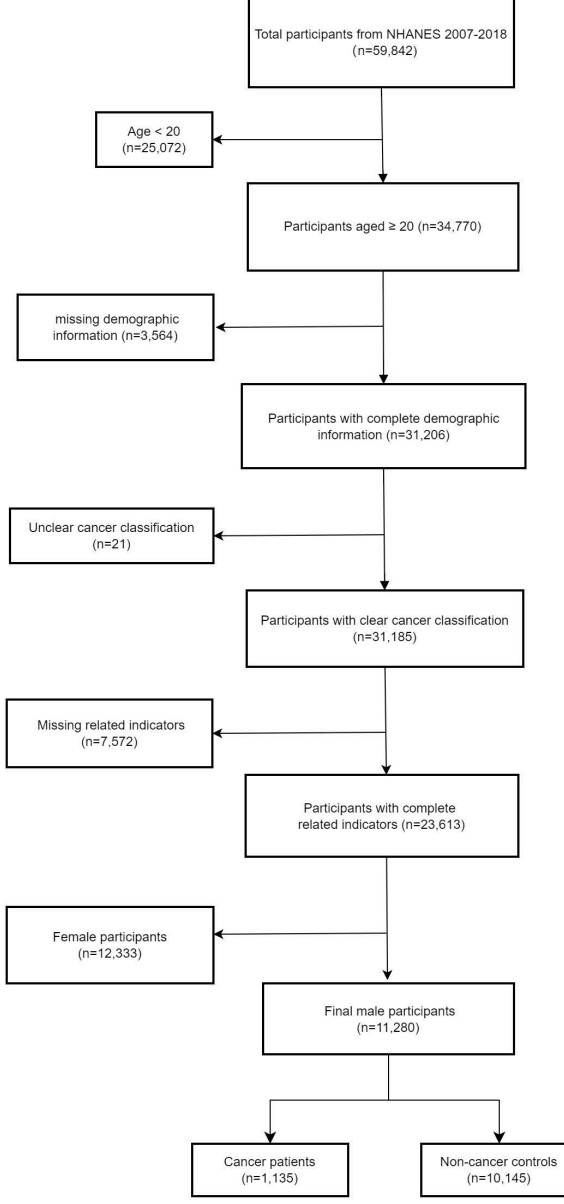

**Fig 1. Flow chart of research object screening.**

executes with high specificity and accuracy. All samples undergo processing under strict quality control conditions, with an intra-laboratory coefficient of variation less than 5%. The laboratory measures testosterone in units of ng/dL and estradiol in pg/mL. All sex hormone measurements are conducted in NHANES morning fasting blood samples to minimize the effects of diurnal variation.

**Calculation methods of inflammatory index.** The ALI was originally developed by Jafri et al. [9] as a prognostic indicator for metastatic non-small cell lung cancer and calculated using the formula: Body Mass Index (BMI, kg/m²) × Serum Albumin (g/dL)/ NLR. This index integrates BMI (reflecting nutritional status), serum albumin (a marker of both nutritional status and systemic inflammation), and NLR (neutrophil-to-lymphocyte ratio, an established inflammatory

marker). ALI's validity has been confirmed in multiple studies across various cancers, including colorectal cancer (Pian et al., 2022) [12] and intrahepatic cholangiocarcinoma (Catalano et al., 2024) [10], with a low ALI associated with poorer survival outcomes. It offers advantages over single markers by incorporating both inflammatory and nutritional dimensions of cancer pathophysiology. In our study, all ALI components were measured according to standardized NHANES protocols using calibrated equipment and automated analyzers with strict quality control standards. BMI data were obtained from NHANES standardized physical examinations, where trained technicians measured height and weight using calibrated equipment. Serum albumin was determined by the bromocresol purple method and expressed in g/dL. Automated hematology analyzers measured neutrophil and lymphocyte counts, with the NLR calculated by dividing the neutrophil count by the lymphocyte count. All measurements were conducted following NHANES standard operating procedures to ensure data accuracy and comparability.

**Selection and measurement of covariates.** Multiple potential confounding factors were considered as covariates. Demographic variables included age (a continuous variable measured in years), race/ethnicity (categorized into Mexican Americans, other Hispanic groups, non-Hispanic whites, non-Hispanic blacks, and other races), education level, and family income-to-poverty ratio (PIR, reflecting socioeconomic status). Moreover, lifestyle factors were primarily assessed through smoking status, with participants classified into five categories: never smokers, former smokers (quit ≥5 years ago), former smokers (quit 1–5 years ago), former smokers (quit <1 year ago), and current smokers. Family secondhand smoke exposure was evaluated based on the SMQFAM module questionnaire. Regarding cancer information, we used the MCQ220 variable (whether a doctor had ever diagnosed cancer) and MCQ230A variable (cancer type) for confirmation and classification. All these covariates were derived from NHANES standardized questionnaires, physical examinations, and laboratory measurements.

**Smoking exposure scoring system.** To comprehensively evaluate the potential impact of nicotine exposure on study results, we referenced the LE8 scoring system and constructed an integrated smoking assessment score based on detailed smoking behavior information from NHANES questionnaires and family secondhand smoke exposure data. The scoring criteria were established as follows: Participants who never smoked or smoked fewer than 100 cigarettes in their lifetime received the highest score of 100 points; those who had quit smoking for 5 years or more were awarded 75 points; individuals who quit smoking 1–5 years ago earned 50 points; those who quit smoking less than 1 year ago were given 25 points; current smokers were assigned 0 points. Additionally, considering the effects of secondhand smoke exposure, 20 points were subtracted from the base score for participants living in households with smokers, with the total score not less than 0. Quit time was calculated by converting the reported time quantity to years based on the corresponding time units. This comprehensive scoring method enables a more accurate quantification and stratified analysis of varying degrees of tobacco exposure, thereby more effectively controlling for this critical confounding factor.

## Statistical analysis

**Descriptive statistics.** To comprehensively understand the research population characteristics and inter-group differences, we employed descriptive statistical methods that considered the NHANES complex sampling design for analyzing all variables. The NHANES standard weighting method was used to construct the complex sampling design object, ensuring national representative estimates, and data weights for 2007–2018 were appropriately adjusted according to NHANES guidelines.

Continuous variables (including age, BMI, albumin, neutrophil count, lymphocyte count, NLR, ALI, smoking score, and PIR) were represented by weighted means and standard deviations (SD). We presented categorical variables (including race/ethnicity, education level, age groups, and PIR groups) using weighted percentages and actual sample sizes (n). Descriptive statistics were displayed for three groups: the entire male sample, male cancer group, and male non-cancer group, accompanied by significance test results.

For continuous variables, we used t-tests that accounted for sampling design to compare differences between cancer and non-cancer groups; categorical variables were compared using chi-square tests considering the sampling design. All statistical analyses incorporated sample weights, stratification, and clustering effects to ensure national representativeness and the accuracy of statistical inferences. Additionally, appropriate categorization was performed on demographic variables (such as age, race, education level, and PIR) to better represent the distribution of population characteristics.

To account for NHANES complex survey design and ensure nationally representative estimates, we applied appropriate sampling weights following NHANES analytical guidelines. Since our analysis combined data from six survey cycles (2007–2018), we created modified sampling weights by dividing the original 2-year examination weights (WTMEC2YR) by 6. For multivariable analyses, we incorporated the stratification variable (SDMVSTRA) and primary sampling unit (SDMVPSU) along with these modified weights using the svydesign function in R with the following specification: study_design <- svydesign(id = ~SDMVPSU, strata = ~SDMVSTRA, weights = ~WTMEC2YR, data = data, nest = TRUE). This approach accounts for the differential selection probabilities, non-response adjustments, and post-stratification alignments to census population totals while properly adjusting standard errors for the complex sampling design.

**Correlation analysis.** The weighted regression models were conducted to evaluate the relationship between sex hormone levels and the ALI, as well as the association of these factors with cancer risk. All analyses considered the NHANES complex sampling design, using the svydesign function from the survey package (https://cran.r-project.org/web/packages/survey/index.html) in R to construct the survey design object, incorporating weight-related information to ensure representation of the American adult male population. When addressing the single Primary Sampling Unit (PSU) stratum issue in the complex sampling design, we applied the "survey.lonely.psu = adjust" option to modify variance estimation, thereby preventing the potential underestimation of standard errors.

In each analysis, three progressively adjusted multivariable models were performed:

(1) Model 1: an unadjusted model;

(2) Model 2: adjusted for age and race/ethnicity;

(3) Model 3: further adjusted for socioeconomic and lifestyle factors, including education level, PIR, and smoking score.

For all outcome variables (including the continuous variable ALI and binary cancer status), we fitted generalized linear models using the svyglm function, and uniformly converted results to odds ratios (OR) with their 95% confidence intervals for reporting. This approach enabled better comparability across different models and facilitated clinical interpretation. Moreover, all statistical tests were conducted using two-sided tests, with P values less than 0.05 considered statistically significant.

To explore nonlinear relationships, we categorized key variables (ALI, testosterone, estradiol) into quartiles, using the svyquantile function to determine quartile points based on weighted distribution. The lowest quartile (Q1) was used as a reference to assess the association of other quartiles with the outcome variable. To verify the robustness of the results, we performed extensive subgroup analyses, stratified by age group (20–39 years, 40–59 years, ≥ 60 years), race/ethnicity, education level, and PIR. In each subgroup, the same three-model analysis strategy was applied, with the subset function used to define specific populations. Given that sample sizes for some subgroups might be limited, we set minimum sample size requirements for each analytical model and carefully considered the precision of estimates during result interpretation.

**Dose-response relationship analysis.** To explore the nonlinear relationship between sex hormone levels and the ALI, we performed dose-response analysis using restricted cubic spline (RCS) method. Using the rms and survey packages in R, RCS models with 5 knots were fitted for each key exposure variable (testosterone and estradiol), considering the NHANES complex sampling design and adjusting for potential confounding factors. For each RCS model, we performed an overall effect test and a nonlinearity test to assess the overall effect of the exposure variable (Overall P) and whether the relationship deviated significantly from linearity (Non-linearity P), respectively.

## Results

### Basic characteristics of study participants

A total of 11,272 American adult male participants were enrolled in this study, with 1,135 (10.1%) cancer patients and 10,137 (89.9%) non-cancer controls. The basic characteristics of the study population were summarized in Table 1.

In terms of age distribution, 35.6% of the total sample were 20–39 years old, 38.3% were 40–59 years old, and 26.1% were 60 years and above. There was a significant difference in age composition between the cancer group and the non-cancer group (P < 0.001). The cancer group was dominated by the elderly (71.3% were 60 years old and above), while the non-cancer group was dominated by the young and middle-aged (78.8% were less than 60 years old). The mean age of the cancer group (68.70 ± 11.85 years) was significantly higher than that of the non-cancer group (48.36 ± 17.00 years).

Regarding race/ethnicity, non-Hispanic whites dominated the population (70.0%), with a higher percentage of 88.3% in the cancer group and 68.1% in the non-cancer group. In contrast, the percentage of minorities in the cancer group was significantly lower than that in the non-cancer group (P < 0.001).

Regarding socioeconomic status, the cancer group demonstrated overall higher education levels and income status compared to the non-cancer group. In the cancer group, 68.3% were found to have high school or higher education, while 60.6% of the non-cancer group were at the same education level (P = 0.001). In terms of the PIR, 66.4% of the cancer group were classified at high-income levels (PIR > 3), compared to 51.5% of the non-cancer group (P < 0.001).

In terms of sex hormone levels, the testosterone levels in the cancer group (377.35 ± 204.26 ng/dL) were significantly lower than that in the non-cancer group (415.02 ± 183.92 ng/dL)(P < 0.001), reflecting the possibility of androgen deficiency in cancer patients. There was no significant difference in estradiol level between the cancer group (24.88 ± 11.56 pg/mL) and the non-cancer group (24.99 ± 9.72 pg/mL) (P = 0.859).

Regarding inflammatory-related indicators, albumin levels in the cancer group (4.19 ± 0.33 g/dL) demonstrated significantly reduced levels relative to the non-cancer group (4.33 ± 0.33 g/dL) (P < 0.001). Neutrophil counts in the cancer group (4.39 ± 1.67 × 10^9/L) were higher compared to the non-cancer group (4.22 ± 1.93 × 10^9/L) (P = 0.003), resulting in a significantly elevated NLR in the cancer group (2.80 ± 1.79 vs 2.19 ± 1.25, P < 0.001). The ALI in the cancer group (64.86 ± 118.61) was lower than in the non-cancer group (72.68 ± 185.25), with the difference approaching statistical significance (P = 0.053).

There was no significant difference in BMI between the two groups (P = 0.430), but the smoking score of the cancer group (72.96 ± 32.33) was higher than that of the non-cancer group (66.27 ± 40.40)(P < 0.001), indicating that the cumulative smoking exposure of cancer patients was more serious.

### Association between sex hormone levels and ALI

We analyzed the association of testosterone and estradiol levels with ALI using weighted multivariate regression models and performed stratified analyses to explore association patterns across different demographic subgroups. The results are presented in Tables 2 and 3.

### Overall analysis results

Table 2 revealed that in all three adjusted models, testosterone demonstrated a significant negative correlation with ALI, while estradiol showed a significant positive correlation with ALI. In the fully adjusted model (Model 3), each 10 ng/dL increase in testosterone level was associated with a 3.0% decrease in ALI (OR=0.970, 95%CI: 0.962–0.978, P < 0.001). In the contrast, each 1 pg/mL increase in estradiol level resulted in a 60.3% increase in ALI (OR=1.603, 95%CI: 1.318–1.949, P < 0.001). These associations were consistently maintained across the three models, indicating that the relationship between sex hormones and ALI remained independent of potential confounding factors.

**Table 1. Characteristics of the study participants: NHANES 2007–2018.**

| Variable | level | Overall (N = 11272) | Cancer (n = 1135) | Non-cancer (n = 10137) | P value |
|---|---|---|---|---|---|
| Age(%) | 20-39 | 3592 (35.6%) | 38 (5.4%) | 3554 (38.9%) | <0.001 |
| | 40-59 | 3687 (38.3%) | 160 (23.3%) | 3527 (39.9%) | |
| | ≥60 | 3993 (26.1%) | 937 (71.3%) | 3056 (21.2%) | |
| Race (%) | Mexican American | 1577 (8.3%) | 50 (1.8%) | 1527 (9.0%) | <0.001 |
| | Other Hispanic | 1016 (5.2%) | 60 (2.1%) | 956 (5.5%) | |
| | Non-Hispanic White | 5156 (70.0%) | 812 (88.3%) | 4344 (68.1%) | |
| | Non-Hispanic Black | 2268 (9.4%) | 169 (4.8%) | 2099 (9.9%) | |
| | Other | 1255 (7.1%) | 44 (3.0%) | 1211 (7.6%) | |
| Education Level (%) | Under high school | 2581 (14.7%) | 210 (10.1%) | 2371 (15.2%) | 0.001 |
| | High school or equivalent | 2701 (24.0%) | 264 (21.5%) | 2437 (24.2%) | |
| | Above high school | 5990 (61.3%) | 661 (68.3%) | 5329 (60.6%) | |
| PIR(%) | <1 | 2122 (12.1%) | 115 (5.7%) | 2007 (12.8%) | <0.001 |
| | 1-3 | 4749 (34.9%) | 476 (27.9%) | 4273 (35.7%) | |
| | >3 | 4401 (53.0%) | 544 (66.4%) | 3857 (51.5%) | |
| Age (mean (±SD)) | | 50.41 (±17.65) | 48.36 (±17.00) | 68.70 (±11.85) | <0.001 |
| Albumin (mean (±SD)) | | 4.31 (±0.33) | 4.33 (±0.33) | 4.19 (±0.33) | <0.001 |
| ALI (mean (±SD)) | | 71.91 (±179.74) | 72.68 (±185.25) | 64.86 (±118.61) | 0.053 |
| BMI(mean (±SD)) | | 28.95 (±6.15) | 28.96 (±6.19) | 28.82 (±5.71) | 0.430 |
| Lymphocyte (mean (±SD)) | | 2.12 (±1.74) | 2.12 (±0.92) | 2.15 (±4.76) | 0.826 |
| Neutrophil (mean (±SD)) | | 4.24 (±1.91) | 4.22 (±1.93) | 4.39 (±1.67) | 0.003 |
| NLR (mean (±SD)) | | 2.25 (±1.32) | 2.19 (±1.25) | 2.80 (±1.79) | <0.001 |
| PIR(mean (±SD)) | | 2.63 (±1.63) | 2.58 (±1.63) | 3.01 (±1.58) | <0.001 |
| Smoke Score(mean (±SD)) | | 66.95 (±39.71) | 66.27 (±40.40) | 72.96 (±32.33) | <0.001 |
| Testosterone(mean (±SD)) | | 411.45(±186.25) | 377.35(±204.26) | 415.02(±183.92) | <0.001 |
| Estradiol(mean (±SD)) | | 24.98(±9.92) | 24.88(±11.56) | 24.99(±9.72) | 0.859 |

**Table 2. Correlation analysis between ALI and sex hormone levels.**

| Predictor Variable | Model 1 (Unadjusted) | | Model 2 (Basic Adjusted) | | Model 3 (Fully Adjusted) | |
|---|---|---|---|---|---|---|
| | OR (95% CI) | P value | OR (95% CI) | P value | OR (95% CI) | P value |
| Testosterone | 0.975 (0.967, 0.982) | <0.001 | 0.969 (0.961, 0.976) | <0.001 | 0.970 (0.962, 0.978) | <0.001 |
| Estradiol | 1.522 (1.236, 1.875) | <0.001 | 1.627 (1.330, 1.991) | <0.001 | 1.603 (1.318, 1.949) | <0.001 |

## Subgroup analysis results

Age Stratification Analysis: The negative correlation between testosterone and ALI was significantly present across all age groups, but appeared most pronounced in the young group (20−39 years) (OR=0.966, 95%CI: 0.953–0.978, P<0.001). Estradiol and ALI showed the strongest positive correlation in the young group (OR=2.008, 95%CI: 1.581–2.549, P<0.001), showed weaker correlation in the middle-aged group (40−59 years) (OR=1.630, 95%CI: 1.041–2.552, P=0.041), and maintained the weakest yet still statistically significant correlation in the elderly group (≥60 years) (OR=1.385, 95%CI: 1.009–1.900, P=0.048).

**Table 3. Correlation analysis between ALI and sex hormone levels in subgroups with different demographic characteristics.**

| Stratification Variable | Predictor Variable | Model 1 (Unadjusted) | | Model 2 (Basic Adjusted) | | Model 3 (Fully Adjusted) | |
|---|---|---|---|---|---|---|---|
| | | OR (95%CI) | P value | OR (95%CI) | P value | OR (95%CI) | P value |
| Age | | | | | | | |
| 20-39 | Testosterone | 0.965 (0.953, 0.978) | <0.001 | 0.964 (0.952, 0.977) | <0.001 | 0.966 (0.953, 0.978) | <0.001 |
| | Estradiol | 2.084 (1.631, 2.662) | <0.001 | 2.033 (1.592, 2.594) | <0.001 | 2.008 (1.581, 2.549) | <0.001 |
| 40-59 | Testosterone | 0.967 (0.953, 0.981) | <0.001 | 0.965 (0.952, 0.979) | <0.001 | 0.966 (0.952, 0.980) | <0.001 |
| | Estradiol | 1.617 (1.036, 2.525) | 0.044 | 1.642 (1.051, 2.566) | 0.037 | 1.630 (1.041, 2.552) | 0.041 |
| ≥60 | Testosterone | 0.982 (0.969, 0.994) | 0.009 | 0.978 (0.966, 0.991) | 0.003 | 0.979 (0.966, 0.991) | 0.004 |
| | Estradiol | 1.311 (0.953, 1.804) | 0.108 | 1.397 (1.018, 1.917) | 0.041 | 1.385 (1.009, 1.900) | 0.048 |
| Race | | | | | | | |
| Mexican American | Testosterone | 0.971 (0.953, 0.989) | 0.005 | 0.967 (0.949, 0.985) | 0.002 | 0.968 (0.951, 0.985) | 0.002 |
| | Estradiol | 1.791 (1.268, 2.528) | 0.004 | 1.796 (1.293, 2.496) | 0.003 | 1.783 (1.295, 2.455) | 0.003 |
| Other Hispanic | Testosterone | 0.961 (0.943, 0.980) | <0.001 | 0.960 (0.942, 0.979) | <0.001 | 0.966 (0.947, 0.985) | 0.002 |
| | Estradiol | 1.415 (0.988, 2.028) | 0.07 | 1.416 (0.999, 2.005) | 0.062 | 1.334 (0.952, 1.870) | 0.109 |
| Non-Hispanic White | Testosterone | 0.976 (0.966, 0.986) | <0.001 | 0.971 (0.961, 0.981) | <0.001 | 0.971 (0.961, 0.981) | <0.001 |
| | Estradiol | 1.313 (0.986, 1.749) | 0.074 | 1.474 (1.113, 1.952) | 0.011 | 1.465 (1.122, 1.912) | 0.01 |
| Non-Hispanic Black | Testosterone | 0.970 (0.952, 0.989) | 0.004 | 0.965 (0.946, 0.984) | 0.001 | 0.975 (0.955, 0.995) | 0.021 |
| | Estradiol | 1.737 (1.229, 2.456) | 0.004 | 1.714 (1.221, 2.407) | 0.004 | 1.513 (1.061, 2.158) | 0.031 |
| Other | Testosterone | 0.972 (0.950, 0.995) | 0.02 | 0.964 (0.941, 0.988) | 0.005 | 0.965 (0.942, 0.989) | 0.006 |
| | Estradiol | 1.403 (0.829, 2.376) | 0.218 | 1.438 (0.845, 2.447) | 0.19 | 1.425 (0.834, 2.435) | 0.207 |
| Education Level | | | | | | | |
| Under high school | Testosterone | 0.976 (0.959, 0.993) | 0.011 | 0.972 (0.956, 0.990) | 0.005 | 0.981 (0.965, 0.998) | 0.036 |
| | Estradiol | 1.596 (1.212, 2.100) | 0.002 | 1.685 (1.253, 2.268) | 0.002 | 1.547 (1.144, 2.092) | 0.009 |
| High school or equivalent | Testosterone | 0.975 (0.960, 0.991) | 0.004 | 0.971 (0.955, 0.986) | <0.001 | 0.973 (0.957, 0.988) | 0.002 |
| | Estradiol | 1.578 (1.080, 2.306) | 0.026 | 1.600 (1.102, 2.325) | 0.016 | 1.547 (1.057, 2.264) | 0.029 |
| Above high school | Testosterone | 0.974 (0.962, 0.987) | <0.001 | 0.967 (0.955, 0.979) | <0.001 | 0.967 (0.955, 0.979) | <0.001 |
| | Estradiol | 1.486 (1.098, 2.011) | 0.016 | 1.630 (1.203, 2.207) | 0.004 | 1.634 (1.213, 2.201) | 0.003 |
| PIR (%) | | | | | | | |
| <1 | Testosterone | 0.977 (0.962, 0.992) | 0.006 | 0.971 (0.956, 0.988) | 0.002 | 0.974 (0.957, 0.991) | 0.007 |
| | Estradiol | 1.418 (1.069, 1.881) | 0.022 | 1.464 (1.090, 1.967) | 0.018 | 1.418 (1.060, 1.896) | 0.026 |
| 1-3 | Testosterone | 0.974 (0.959, 0.990) | 0.003 | 0.965 (0.949, 0.981) | <0.001 | 0.967 (0.951, 0.983) | <0.001 |
| | Estradiol | 1.462 (1.066, 2.006) | 0.026 | 1.606 (1.181, 2.185) | 0.006 | 1.536 (1.132, 2.084) | 0.011 |
| >3 | Testosterone | 0.974 (0.963, 0.985) | <0.001 | 0.971 (0.962, 0.981) | <0.001 | 0.971 (0.961, 0.981) | <0.001 |
| | Estradiol | 1.594 (1.178, 2.158) | 0.005 | 1.695 (1.279, 2.245) | 0.001 | 1.704 (1.302, 2.230) | <0.001 |

Across ethnic subgroups, A significant negative correlation between testosterone and ALI existed across all racial/ethnic groups, with all groups showing relatively consistent effect sizes (OR range: 0.965–0.975). Meanwhile, estradiol showed a positive association with ALI that varied by ethnicity: strongest in Mexican Americans (OR=1.783, 95%CI: 1.295–2.455, P=0.003), substantial in non-Hispanic blacks (OR=1.513, 95%CI: 1.061–2.158, P=0.031) and non-Hispanic whites (OR=1.465, 95%CI: 1.122–1.912, P=0.010), but not statistically significant among other Hispanic groups and racial categories.

Education Level Stratification Analysis: The negative correlation between testosterone and ALI was significantly present across all education level groups, but appeared more evident in the high education level group (OR=0.967, 95%CI: 0.955–0.979, P<0.001). A significant negative correlation between estradiol and ALI existed across all education level groups, with all groups showing relatively consistent effect sizes (OR range: 1.547–1.634).

Income Level Stratification Analysis: Testosterone consistently showed an inverse relationship with ALI across all income level group, with similar effect sizes (OR range: 0.967–0.974). The positive correlation between estradiol and ALI was also significant across subgroups, but demonstrated the strongest effect in the high-income group (PIR > 3) (OR=1.704, 95%CI: 1.302–2.230, P < 0.001).

### Sex hormones and ALI: Cancer status stratified analysis

To explore whether cancer status modulated the association between sex hormones and ALI, we performed stratified analyses according to cancer status. The results were presented in Table 4 and S1 and S2 Tables.

**Modulating effect of cancer status on the association between sex hormones and ALI.** The significant negative correlation between testosterone and ALI was only observed in non-cancer populations (OR=0.97, 95%CI: 0.96–0.98, P < 0.001). However, in cancer patients, the direction was consistent but did not reach statistical significance (OR=0.98, 95%CI: 0.94–1.03, P = 0.412). Similarly, the positive association between estradiol and ALI was only significant among individuals without cancer (OR=1.64, 95%CI: 1.39–1.94, P < 0.001), while no significant association was found in cancer patients (OR=1.27, 95%CI: 0.34–4.83, P = 0.725). These findings suggest that cancer status may alter the association patterns between sex hormones and inflammatory markers.

**Interaction between cancer status and age.** The age-stratified analyses (S1 and S2 Tables) revealed more complex association patterns. For testosterone, a significant negative correlation with ALI was consistently maintained across all age groups in non-cancer individuals. However, the situation differed for cancer patients, where only the elderly cohort (≥60 years) exhibited a significant inverse relationship (OR=0.96, 95%CI: 0.92–0.99, P = 0.020).

Regarding estradiol, non-cancer populations demonstrated an age-dependent attenuation of its positive association with ALI. Furthermore, the effect was strongest in the youngest group (OR=2.07, 95%CI: 1.64–2.61, P < 0.001), which was subsequently diminished in middle-aged participants (OR=1.80, 95%CI: 1.31–2.46, P = 0.001), and ultimately lost significance among the elderly (OR=1.11, 95%CI: 0.89–1.38, P = 0.372). However, in cancer patients, the association between estradiol and ALI did not reach a statistically significant level in all age groups.

**Interaction between cancer status and race/ethnicity.** In the non-cancer population, the inverse association between testosterone and ALI was significant in all racial/ethnic groups, with similar effect sizes (OR=0.97, 95%CI: 0.96–0.98). However, no significant association was observed for all ethnic groups in cancer patients.

The association between estradiol and ALI was more pronounced by race in the non-cancer population, being significant in Mexican Americans (OR=1.78, P = 0.005) and non-Hispanic whites (OR=1.52, P = 0.001), but marginal in non-Hispanic blacks (OR=1.42, P = 0.057). Among cancer patients, none of the ethnic groups showed a significant association.

**Interaction between cancer status and socioeconomic status.** Regarding education level, a significant negative correlation between testosterone and ALI was detected in the high school and above education groups (P < 0.01) among non-cancer individuals, while the low education group did not reach statistical significance (OR=0.98, P = 0.061). No significant associations were found across any education level groups among cancer patients. The relationship between estradiol and ALI showed minimal variation across education levels in non-cancer populations, with all education groups

Table 4. Correlation analysis between ALI and sex hormone levels in different cancer states.

| Predictor Variable | Cancer Status | Model 1 (Unadjusted) | | | Model 2 (Basic Adjusted) | | | Model 3 (Fully Adjusted) | | |
|---|---|---|---|---|---|---|---|---|---|---|
| | | OR | 95% CI | P value | OR | 95% CI | P value | OR | 95% CI | P value |
| Testosterone | Non-Cancer | 0.97 | 0.97-0.98 | <0.001 | 0.97 | 0.96-0.98 | <0.001 | 0.97 | 0.96-0.98 | <0.001 |
| | Cancer | 0.99 | 0.94-1.04 | 0.601 | 0.98 | 0.94-1.03 | 0.511 | 0.98 | 0.94-1.03 | 0.412 |
| Estradiol | Non-Cancer | 1.58 | 1.33-1.88 | <0.001 | 1.68 | 1.43-1.97 | <0.001 | 1.64 | 1.39-1.94 | <0.001 |
| | Cancer | 1.13 | 0.23-5.65 | 0.88 | 1.15 | 0.24-5.52 | 0.865 | 1.27 | 0.34-4.83 | 0.725 |

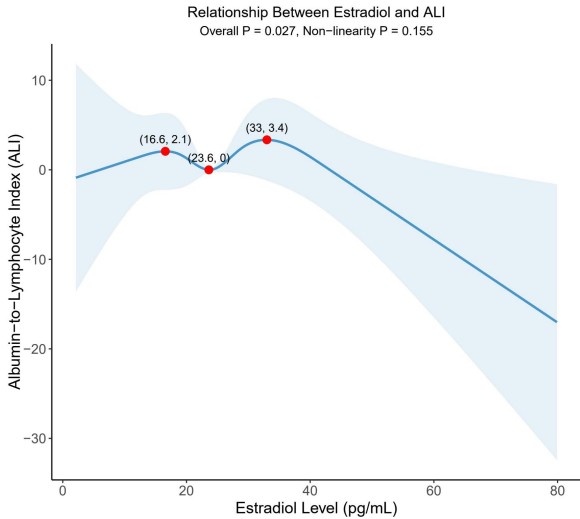

exhibiting significant or nearly significant positive correlations. Cancer patients demonstrated no significant associations across all education level categories.

For income level, testosterone negatively correlated with ALI in a significant manner across all income groups (all P < 0.05) within the non-cancer population. However, cancer patients showed no significant associations in any income bracket. The positive correlation between estradiol and ALI was significant across all income levels in non-cancer individuals (all P < 0.05), with the strongest effect observed in the high-income group (OR=1.89, 95%CI: 1.50–2.39, P < 0.001). No significant associations were observed in any income level group among cancer patients.

Overall, the results of our stratified analysis showed that the association between sex hormones and ALI was significantly different in cancer patients and non-cancer population. The association between sex hormones and ALI was more consistent and significant in non-cancer population, while in cancer patients, these associations became unstable and more heterogeneous. This difference was further moderated by age, race/ethnicity, and socioeconomic status.

**Nonlinear association analysis of sex hormones and ALI.** To delve into the potential non-linear relationship between sex hormones and ALI, RCS method was used for analyses. As shown in S1 Fig, the relationship between testosterone and ALI showed a certain nonlinear trend, but did not reach a statistically significant level (overall P = 0.222, nonlinear P = 0.385). In contrast, estradiol demonstrated a markedly more pronounced non-linear association with ALI, achieving statistical significance (overall P = 0.027), although the nonlinear term did not reach significant level (nonlinear P = 0.155). Furthermore, the non-linear relationship between estradiol and ALI showed different association patterns at three key inflection-points (Fig 2):

(1) Before the first inflection-point (16.6 pg/mL), estradiol and ALI showed a positive correlation, and ALI value increased to 2.1 with the increase of estradiol.

(2) Within the interval of 16.6–23.6 pg/mL, the correlation turned negative, and the ALI value decreased to 0.

(3) Between 23.6 and 33.0 pg/mL, the relationship turned to positive again, and the ALI value increased to 3.4.

(4) Beyond 33.0 pg/mL, the curve gradually flattened out.

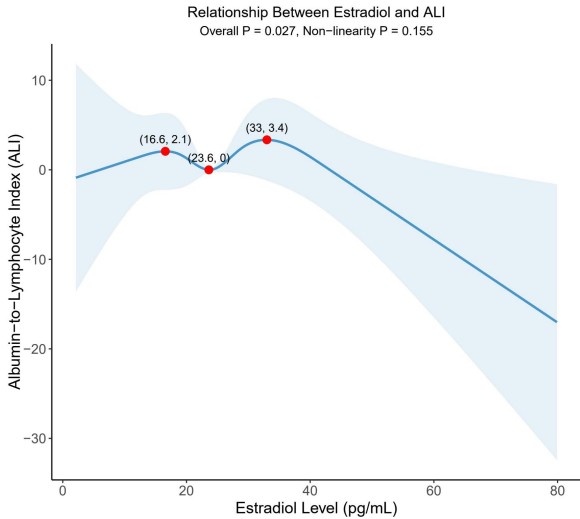

**Fig 2. Restricted cubic spline curves of nonlinear relationship between estradiol and ALI.**

This intricate associative pattern illuminated the potential threshold and concentration-dependent effects of changes in estradiol levels on ALI. Notably, the examined concentration range encompassed the estradiol distribution of the vast majority of individuals within the study population.

**Association analysis of sex hormones, ALI, and cancer risk.** In the continuous variable analysis (Table 5), none of ALI, testosterone, none of ALI, testosterone, and estradiol were significantly and independently associated with cancer risk. ALI demonstrated non-significant significant in both the unadjusted model (OR=1.005, 95%CI: 0.997–1.013, P = 0.216) and the fully adjusted model (OR=1.000, P = 0.738). Testosterone was weakly inversely associated with cancer risk in the unadjusted model (OR=0.999, P = 0.003), but this association disappeared after adjustment for confounding factors (fully adjusted model: OR=1.000, P = 0.208). Estradiol failed to exhibit any significant association with cancer risk across all model specifications (fully adjusted model: OR=1.002, 95%CI: 0.990–1.014, P = 0.759).

The analysis of quartile grouping in Supplementary S3–S5 Tables showed a similar pattern. In the unadjusted models, higher levels of ALI (Q4 vs Q1: OR=0.37, P < 0.001) and testosterone (Q4 vs Q1: OR=0.664, P = 0.0238) were significantly associated with reduced cancer risk, with pronounced linear trends (P for trend<0.001 and P = 0.0167). However, these associations disappeared after adjusting for age, race and other confounding factors. Estradiol levels were not significantly associated with cancer risk in any of the models. Notably, even in the unadjusted model there was only a slight non-significant trend toward increased risk (Q4 vs Q1: OR=1.287, P = 0.19; P for trend = 0.0806).

## Discussion

This study is the first to systematically evaluate and identify distinct association patterns between sex hormones and ALI in both general and cancer populations. In the general population, testosterone levels demonstrated a significant negative correlation with ALI, with each 10 ng/dL increase in testosterone levels associated with a 3.0% decrease in ALI. In contrast, a strong positive correlation was observed between estradiol levels and ALI, with ALI being increased by 60.3% for each 1 pg/mL increase in estradiol levels. This association remained stable across three progressively adjusted models, indicating that the relationship between sex hormones and ALI is independent of potential confounding factors. These findings were highly consistent with existing literature. Regarding the anti-inflammatory effects of testosterone, it was reported by Malkin et al. [20] that levels of pro-inflammatory cytokines such as TNF-α, IL-1β, and IL-6 in males could be reduced by testosterone replacement therapy; Lanser et al. [21] found that men with low testosterone exhibited higher levels of CRP and IL-6. This anti-inflammatory effect might be achieved through multiple mechanisms, including inhibiting the NF-κB pathway, reducing pro-inflammatory cytokine production, enhancing anti-inflammatory cytokine release, and modulating immune cell function [22]. Our research further confirmed the widespread existence of this relationship at the population level and, for the first time, established its association with ALI, a comprehensive inflammatory indicator.

A positive correlation between estradiol and ALI was supported by emerging evidence, although relatively less studied in males. Tsilidis et al. [23] found that higher estradiol levels in males correlated with elevated C-reactive protein levels; Similar patterns were observed by Dubey et al. [24]. These gender differences might originate from unique estrogen signaling pathway activation in males or relate to differential estrogen receptor expression in male tissues. In males, estradiol is primarily derived from testosterone aromatization, with concentrations significantly lower than those in females. These

**Table 5. Results of weighted logistic regression analysis of sex hormones, ALI, and cancer risk in men.**

| Variable | Model 1 (Unadjusted) | | Model 2 (Basic Adjusted) | | Model 3 (Fully Adjusted) | |
|---|---|---|---|---|---|---|
| | OR (95% CI) | P value | OR (95% CI) | P value | OR (95% CI) | P value |
| ALI | 1.005 (0.997-1.013) | 0.216 | 1.000 (1.000-1.000) | 0.652 | 1.000 (1.000-1.000) | 0.738 |
| Testosterone | 0.999 (0.998-1.000) | 0.003 | 1.000 (1.000-1.000) | 0.293 | 1.000 (1.000-1.000) | 0.208 |
| Estradiol | 1.007 (0.997-1.016) | 0.173 | 1.002 (0.990-1.013) | 0.784 | 1.002 (0.990-1.014) | 0.759 |

lower concentrations might lead to distinct immunomodulatory effects through different receptor subtypes and signaling pathways [25].

Further subgroup analyses revealed that the negative correlation between testosterone and ALI was most pronounced in the young population (20–39 years), significantly present across all racial/ethnic groups, with relatively consistent effect sizes. However, the strongest positive correlation between estradiol and ALI was observed in the younger group, weakening with age; the effect was most prominent in Mexican Americans and non-Hispanic Blacks. These findings might reflect differential sensitivity of different populations to the inflammatory regulatory effects of sex hormones. Higher sensitivity to sex hormones' anti-inflammatory effects in young individuals could be associated with age-related changes in hormone receptor expression or differences in baseline inflammatory levels. Maggio et al. [26] found that the association between decreased testosterone levels and increased inflammatory markers gradually weakens with age, consistent with our results. Additionally, racial differences might be related to genetic backgrounds, lifestyles, and environmental factors. Velásquez-Mieyer et al. [27] observed differences in hormone-inflammation relationships between African Americans and Whites, potentially linked to androgen receptor gene polymorphisms.

However, the most critical finding of this study is that cancer status significantly modulates the association between sex hormones and ALI. Statistically significant negative correlations between testosterone and estradiol with ALI were observed only in non-cancer populations, while no significant associations were found in cancer patients. Furthermore, subgroup analyses revealed that beyond the significant association between testosterone and ALI observed in the elderly group, this pattern broadly disappeared across stratifications by age, race/ethnicity, and socioeconomic factors in cancer patients. This discovery represents a novel scientific insight that may reflect profound alterations in the body's inflammatory regulatory network induced by cancer. First, tumor cells can produce multiple inflammatory factors [28], potentially masking the regulatory effects of sex hormones; second, changes in sex hormone receptor expression or alterations in signaling pathway sensitivity may be induced by cancer [29]; additionally, cancer treatments (such as chemotherapy, radiotherapy, or surgery) could also influence the relationship between sex hormones and inflammation [30]. Although the studies of Winther-Larsen et al. [31] and Huai et al. [32] focused on the relationship between inflammation and cancer, they rarely considered the regulatory role of sex hormones, and our study fills this knowledge gap.

The significant association between testosterone and ALI observed in the elderly group of cancer patients may indicate a unique modulatory effect of age on the sex hormone-inflammation relationship. Older cancer patients might be characterized by more pronounced hormone deficiency and higher baseline inflammatory levels, making the sex hormone-inflammation relationship more readily detectable [26]. Additionally, variations in treatment protocols across different age groups of cancer patients may also affect this relationship [33].

We finally found that neither sex hormones nor ALI were significantly and independently associated with cancer risk after adjusting for confounding factors. This result must be interpreted against the backdrop of important methodological differences with previous research. Initially, ALI was proposed by Jafri et al. [9] and subsequently used primarily as a prognostic indicator for post-cancer surgery. Moreover, Catalano et al. [10], Liu et al. [11], and Pian et al. [12] focused on the value of ALI in predicting survival after cancer treatment, rather than the cancer risk association we investigated.

The fundamental difference lies in the fact that post-surgical ALI reflects the inflammatory state of patients already diagnosed and treated for cancer, whereas our study examined the cross-sectional association between ALI and cancer status in the general population, more closely approximating the "pre-surgical" or diagnostic state.

Secondly, several reasons might account for the inconsistency with other research results: Watts et al. [34] found that low testosterone was associated with increased prostate cancer risk, but their study was specific to prostate cancer, whereas our study included all cancer types; the independent associations of inflammatory markers with cancer risk reported by Demb et al. [35] were mainly based on prospective cohort studies that were better able to establish time-series relationships. Our findings align with the meta-analysis by Roddam et al. [36], which found no significant association between testosterone and overall cancer risk after adjusting for factors like age. Additionally, cancer patients in our study were at various diagnostic and

treatment stages, rather than being assessed at a specific time point (such as pre-surgery). The patients' sex hormone levels and inflammatory states might have already been altered by the process of cancer diagnosis and treatment [32,37], which fundamentally differs from studies specifically focusing on pre- or post-surgical time points.

This study represents the first large-scale population research to systematically explore the association between sex hormones and ALI. By combining three indicators—neutrophils, lymphocytes, and albumin—we reflected both innate and adaptive immune responses as well as nutritional status, thus demonstrating a new metric with more comprehensive value than single inflammatory markers. Additionally, we revealed how cancer status modulates the association between sex hormones and inflammation; however, several important limitations of this work need to be recognized. The cross-sectional study design is the main limitation, preventing us from determining the temporal relationship or causal connection between sex hormones and ALI. Additionally, while the NHANES database provides current cancer status, it lacks critical details such as cancer type, stage, diagnosis timing, and treatment information, which restricts our ability to analyze cancer-specific effects. Another limitation is that sex hormone measurements are only conducted at a single time point, unable to reflect diurnal fluctuations and long-term changes.

These limitations suggest several directions for future research. Prospective cohort studies should be conducted to establish causal relationships. Research should also differentiate between cancer types and treatment protocols, explore underlying biological mechanisms, and evaluate the potential applications of sex hormones and ALI in cancer management. Furthermore, cross-ethnic and cross-cultural studies would help verify the universality of these associations and contribute to the development of personalized medicine approaches.

## Conclusion

This study is the first to systematically investigate the association between male sex hormone levels and ALI, as well as the regulatory role of cancer status. Based on large sample data representing the adult male population in the United States, we found that testosterone was significantly negatively correlated with ALI, while estradiol showed a positive correlation with ALI. These associations remained robust after adjusting for demographic and clinical characteristics. More importantly, these associations generally disappeared in cancer patients, indicating that cancer status profoundly alters the relationship between sex hormones and inflammation. Additionally, a non-linear relationship was observed between estradiol and ALI, presenting multiple inflection points and revealing that estradiol may have concentration-dependent effects. These findings provide new perspectives on how sex hormones differentially regulate inflammatory responses in healthy and disease states, establishing an important foundation for understanding hormone-inflammation interactions in male health, inflammation-related diseases, and cancer.

## Supporting information

**S1 Fig. Restricted cubic spline curves for the nonlinear relationship between testosterone and ALI.**
(TIF)

**S1 Table. Correlation analysis between testosterone levels and ALI in different demographic subgroups under different cancer status.**
(XLSX)

**S2 Table. Correlation analysis between estradiol level and ALI in different demographic subgroups under different cancer status.**
(XLSX)

**S3 Table. Association Between ALI Quartiles and Cancer Risk.**
(XLSX)

**S4 Table. Testosterone Quartiles Correlating with Cancer Risk.**
(XLSX)

**S5 Table. Estradiol Quartile Distribution and Its Cancer Risk Implications.**
(XLSX)

## Acknowledgments

We appreciated all the fellows in our department for their excellent work.

## Author contributions

**Conceptualization:** Wenyao Xie, Dan Zhao, Yicong Zhou.

**Data curation:** Wenyao Xie, Zhenjun Zhang, Dan Zhao, Ya Zhang, Caiting Feng, Yicong Zhou.

**Formal analysis:** Wenyao Xie, Zhenjun Zhang, Dan Zhao, Ya Zhang, Caiting Feng.

**Funding acquisition:** Yicong Zhou.

**Investigation:** Wenyao Xie, Zhenjun Zhang.

**Methodology:** Wenyao Xie, Zhenjun Zhang, Dan Zhao, Ya Zhang, Caiting Feng, Yicong Zhou.

**Project administration:** Yicong Zhou.

**Resources:** Yicong Zhou.

**Software:** Wenyao Xie, Zhenjun Zhang, Dan Zhao, Ya Zhang, Caiting Feng.

**Supervision:** Yicong Zhou.

**Validation:** Wenyao Xie, Dan Zhao, Ya Zhang, Caiting Feng, Yicong Zhou.

**Visualization:** Wenyao Xie, Zhenjun Zhang, Dan Zhao, Ya Zhang, Caiting Feng.

**Writing – original draft:** Wenyao Xie, Zhenjun Zhang, Dan Zhao, Ya Zhang, Caiting Feng.

**Writing – review & editing:** Wenyao Xie, Yicong Zhou.

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
