## [Decision Letter · Decision Letter 0]

Dear Dr. Zhou,

Thank you for submitting your manuscript to PLOS ONE. After careful consideration, we feel that it has merit but does not fully meet PLOS ONE’s publication criteria as it currently stands. Therefore, we invite you to submit a revised version of the manuscript that addresses the points raised during the review process.

Your study on the relationship between sex hormones and inflammation in cancer and non-cancer populations using NHANES data is thoughtful and well-executed. To enhance the clarity and readability of the manuscript, please address the following:

• Consider shortening the title for clarity.

• Revise the abstract to follow a structured format (Background, Methods, Results, Conclusion).

• Clearly define how cancer status was determined in NHANES and provide detailed explanation of how sampling weights across survey cycles were applied in the analysis.

• Make minor grammatical and language edits to improve sentence clarity and overall flow.

We look forward to receiving your revised manuscript.

Kind regards,

Akingbolabo Daniel Ogunlakin, Phd

Academic Editor

PLOS ONE

Journal Requirements:

Reviewers' comments:

Reviewer's Responses to Questions

**Comments to the Author**

1. Is the manuscript technically sound, and do the data support the conclusions?

Reviewer #1: Yes

Reviewer #2: Yes

2. Has the statistical analysis been performed appropriately and rigorously?

Reviewer #1: Yes

Reviewer #2: Yes

3. Have the authors made all data underlying the findings in their manuscript fully available?

Reviewer #1: Yes

Reviewer #2: Yes

4. Is the manuscript presented in an intelligible fashion and written in standard English?

Reviewer #1: Yes

Reviewer #2: Yes

Reviewer #1: The study is a well-executed and impactful contribution that provides a compelling exploration between testosterone, inflammation and advanced lung cancer using a robust dataset from NHANES. The research is methodologically sound and clinically relevant, providing insights into sex hormone dynamics in cancer progression. Below are minor suggestions for improvement. Keywords should not repeat words in the title of the manuscript.

The term Advanced lung cancer index is ambiguous. Authors should explain explicitly how this index was constructed and justify its validity. A detailed explanation against established standards would improve the study.

Reviewer #2: Your study on the relationship between sex hormones and inflammation in cancer and non-cancer populations using NHANES data is thoughtful and well-executed. To enhance the clarity and readability of the manuscript, please address the following:

• Consider shortening the title for clarity.

• Revise the abstract to follow a structured format (Background, Methods, Results, Conclusion).

• Clearly define how cancer status was determined in NHANES and provide detailed explanation of how sampling weights across survey cycles were applied in the analysis.

• Make minor grammatical and language edits to improve sentence clarity and overall flow.

Thank you.

**Do you want your identity to be public for this peer review?** For information about this choice, including consent withdrawal, please see our Privacy Policy

Reviewer #1: No

Reviewer #2: No

---

## [Author Response · Author response to Decision Letter 1]

12 May 2025

Dear Reviewer 1:

We appreciate your careful review of our manuscript and believe that the changes made in response to your suggestions have significantly strengthened the paper. Your expertise and attention to detail have helped us enhance both the technical accuracy and accessibility of our research for readers. Thank you again for your contribution to improving our work. We hope that the revised manuscript now meets your expectations.

1. Keywords should not repeat words in the title of the manuscript.

Thank you for this important formatting suggestion. We agree that keywords should complement the title rather than repeat terms already present in it. Based on your recommendation, we have revised our keywords to avoid duplication with the title while still accurately representing the core concepts of our research. The revised keywords are: Androgen, Immunomodulation, Systemic inflammation, Neutrophil-to-lymphocyte ratio, Albumin, Gender differences.是

2. The term Advanced lung cancer index is ambiguous. Authors should explain explicitly how this index was constructed and justify its validity. A detailed explanation against established standards would improve the study.

Thank you for this valuable comment highlighting the need for greater clarity regarding the Advanced Lung Cancer Inflammation Index (ALI). We agree that a more detailed explanation would strengthen our manuscript. We have expanded our Methods section to include a more comprehensive explanation of ALI:

The ALI was originally developed by Jafri et al. [9] as a prognostic indicator for metastatic non-small cell lung cancer and calculated using the formula: Body Mass Index (BMI, kg/m²) × Serum Albumin (g/dL) / NLR. This index integrates BMI (reflecting nutritional status), serum albumin (a marker of both nutritional status and systemic inflammation), and NLR (neutrophil-to-lymphocyte ratio, an established inflammatory marker). ALI's validity has been confirmed in multiple studies across various cancers, including colorectal cancer (Pian et al., 2022)[12] and intrahepatic cholangiocarcinoma (Catalano et al., 2024)[10], with a low ALI associated with poorer survival outcomes. It offers advantages over single markers by incorporating both inflammatory and nutritional dimensions of cancer pathophysiology. In our study, all ALI components were measured according to standardized NHANES protocols using calibrated equipment and automated analyzers with strict quality control standards.

Dear Reviewer 2:

We sincerely thank you for the careful review and valuable suggestions on our manuscript. Each point raised is of great value in improving the quality of our paper, and we have carefully considered and made corresponding revisions to address all your concerns. We particularly appreciate your attention to the overall framework of our paper, which has significantly enhanced the quality of our work. Your expert insights have helped us identify key areas for improvement in our manuscript, allowing us to refine our research from a more comprehensive and rigorous perspective. In response to each specific issue you raised, we have made detailed modifications throughout the manuscript. All changes have been highlighted in yellow in the revised manuscript for your easy identification.

1. Consider shortening the title for clarity.

Thank you for your thoughtful feedback on our manuscript. We agree that a more concise title would enhance clarity and readability. Following your suggestion, we have revised our title to be more focused while maintaining the core elements of our research. The revised title is "Cancer Disrupts Sex Hormone-Inflammation Relationships: Analysis of ALI in Males from NHANES 2007-2018". This shortened title more efficiently communicates our study's primary finding (the disruption of hormone-inflammation relationships in cancer) while maintaining essential information about the dataset and methodology. The acronym ALI (Advanced Lung Cancer Inflammation Index) is widely recognized in the field and will be fully defined in the abstract and introduction, allowing us to create a more streamlined title without sacrificing clarity.

2. Revise the abstract to follow a structured format (Background, Methods, Results, Conclusion).

Thank you for your insightful feedback on our manuscript. We appreciate your attention to detail and constructive suggestions for improving our work. We have revised the abstract to follow the structured format as suggested. The abstract now includes clearly labeled sections for Background, Methods, Results, and Conclusion.

3. Clearly define how cancer status was determined in NHANES and provide detailed explanation of how sampling weights across survey cycles were applied in the analysis.

We sincerely appreciate your thoughtful review and the opportunity to clarify important methodological aspects of our study. We have expanded our Methods section to provide more comprehensive information on both cancer status determination and sampling weight application: "Cancer status in our study was determined using the NHANES Medical Conditions Questionnaire (MCQ), which collected self-reported data on medical conditions through personal interviews conducted by trained NHANES staff. Specifically, participants were asked "Has a doctor or other health professional ever told you that you had cancer or a malignancy of any kind?" (variable MCQ220). Those responding "Yes" were classified as having a cancer history, while those responding "No" formed our non-cancer group. Participants with "Don't know" or refused responses were excluded from our analysis. For participants confirming a cancer history, additional information was collected regarding cancer type and age at diagnosis. We excluded individuals with lung cancer to avoid potential confounding with our primary outcome measure."

“To account for NHANES complex survey design and ensure nationally representative estimates, we applied appropriate sampling weights following NHANES analytical guidelines. Since our analysis combined data from six survey cycles (2007-2018), we created modified sampling weights by dividing the original 2-year examination weights (WTMEC2YR) by 6. For multivariable analyses, we incorporated the stratification variable (SDMVSTRA) and primary sampling unit (SDMVPSU) along with these modified weights using the svydesign function in R with the following specification: study_design <- svydesign(id = ~SDMVPSU, strata = ~SDMVSTRA, weights = ~WTMEC2YR, data = data, nest = TRUE). This approach accounts for the differential selection probabilities, non-response adjustments, and post-stratification alignments to census population totals while properly adjusting standard errors for the complex sampling design.”

4. Make minor grammatical and language edits to improve sentence clarity and overall flow. We have completed a thorough review of the manuscript and made all necessary grammatical and language edits to improve sentence clarity and overall flow. We carefully addressed each instance of awkward phrasing, refined transition sentences between sections, standardized terminology throughout the text, and ensured consistent verb tense usage. These revisions have enhanced the readability and precision of our work while maintaining the scientific integrity of our findings. We sincerely appreciate the reviewer's attention to these important details, as clear scientific communication is essential for effectively sharing our research contributions with the scientific community. We extend our heartfelt thanks for this valuable feedback, and if there are any remaining concerns about the language or presentation, we would be more than happy to address them in further revisions.

---

## [Editor Report · Decision Letter 1]

Dear Dr. Zhou,

Thank you for submitting your manuscript to PLOS ONE. After careful consideration, we feel that it has merit but does not fully meet PLOS ONE’s publication criteria as it currently stands. Therefore, we invite you to submit a revised version of the manuscript that addresses the points raised during the review process.

Your study on the relationship between sex hormones and inflammation in cancer and non-cancer populations using NHANES data is thoughtful and well-executed. To enhance the clarity and readability of the manuscript, please address the following:

• Consider shortening the title for clarity.

• Revise the abstract to follow a structured format (Background, Methods, Results, Conclusion).

• Clearly define how cancer status was determined in NHANES and provide detailed explanation of how sampling weights across survey cycles were applied in the analysis.

• Make minor grammatical and language edits to improve sentence clarity and overall flow.

We look forward to receiving your revised manuscript.

Kind regards,

Akingbolabo Daniel Ogunlakin, Phd

Academic Editor

PLOS ONE

Journal Requirements:

Additional Editor Comments:

Your study on the relationship between sex hormones and inflammation in cancer and non-cancer populations using NHANES data is thoughtful and well-executed. To enhance the clarity and readability of the manuscript, please address the following:

• Consider shortening the title for clarity.

• Revise the abstract to follow a structured format (Background, Methods, Results, Conclusion).

• Clearly define how cancer status was determined in NHANES and provide detailed explanation of how sampling weights across survey cycles were applied in the analysis.

• Make minor grammatical and language edits to improve sentence clarity and overall flow.

Thank you.

---

## [Author Response · Author response to Decision Letter 2]

15 May 2025

·Dear Reviewer 1:

We appreciate your careful review of our manuscript and believe that the changes made in response to your suggestions have significantly strengthened the paper. Your expertise and attention to detail have helped us enhance both the technical accuracy and accessibility of our research for readers. Thank you again for your contribution to improving our work. We hope that the revised manuscript now meets your expectations.

1. Keywords should not repeat words in the title of the manuscript.

Thank you for this important formatting suggestion. We agree that keywords should complement the title rather than repeat terms already present in it. Based on your recommendation, we have revised our keywords to avoid duplication with the title while still accurately representing the core concepts of our research. The revised keywords are: Androgen, Immunomodulation, Systemic inflammation, Neutrophil-to-lymphocyte ratio, Albumin, Gender differences.是

2. The term Advanced lung cancer index is ambiguous. Authors should explain explicitly how this index was constructed and justify its validity. A detailed explanation against established standards would improve the study.

Thank you for this valuable comment highlighting the need for greater clarity regarding the Advanced Lung Cancer Inflammation Index (ALI). We agree that a more detailed explanation would strengthen our manuscript. We have expanded our Methods section to include a more comprehensive explanation of ALI:

The ALI was originally developed by Jafri et al. [9] as a prognostic indicator for metastatic non-small cell lung cancer and calculated using the formula: Body Mass Index (BMI, kg/m²) × Serum Albumin (g/dL) / NLR. This index integrates BMI (reflecting nutritional status), serum albumin (a marker of both nutritional status and systemic inflammation), and NLR (neutrophil-to-lymphocyte ratio, an established inflammatory marker). ALI's validity has been confirmed in multiple studies across various cancers, including colorectal cancer (Pian et al., 2022)[12] and intrahepatic cholangiocarcinoma (Catalano et al., 2024)[10], with a low ALI associated with poorer survival outcomes. It offers advantages over single markers by incorporating both inflammatory and nutritional dimensions of cancer pathophysiology. In our study, all ALI components were measured according to standardized NHANES protocols using calibrated equipment and automated analyzers with strict quality control standards.

Dear Reviewer 2:

We sincerely thank you for the careful review and valuable suggestions on our manuscript. Each point raised is of great value in improving the quality of our paper, and we have carefully considered and made corresponding revisions to address all your concerns. We particularly appreciate your attention to the overall framework of our paper, which has significantly enhanced the quality of our work. Your expert insights have helped us identify key areas for improvement in our manuscript, allowing us to refine our research from a more comprehensive and rigorous perspective. In response to each specific issue you raised, we have made detailed modifications throughout the manuscript. All changes have been highlighted in yellow in the revised manuscript for your easy identification.

1. Consider shortening the title for clarity.

Thank you for your thoughtful feedback on our manuscript. We agree that a more concise title would enhance clarity and readability. Following your suggestion, we have revised our title to be more focused while maintaining the core elements of our research. The revised title is "Cancer Disrupts Sex Hormone-Inflammation Relationships: Analysis of ALI in Males from NHANES 2007-2018". This shortened title more efficiently communicates our study's primary finding (the disruption of hormone-inflammation relationships in cancer) while maintaining essential information about the dataset and methodology. The acronym ALI (Advanced Lung Cancer Inflammation Index) is widely recognized in the field and will be fully defined in the abstract and introduction, allowing us to create a more streamlined title without sacrificing clarity.

2. Revise the abstract to follow a structured format (Background, Methods, Results, Conclusion).

Thank you for your insightful feedback on our manuscript. We appreciate your attention to detail and constructive suggestions for improving our work. We have revised the abstract to follow the structured format as suggested. The abstract now includes clearly labeled sections for Background, Methods, Results, and Conclusion.

3. Clearly define how cancer status was determined in NHANES and provide detailed explanation of how sampling weights across survey cycles were applied in the analysis.

We sincerely appreciate your thoughtful review and the opportunity to clarify important methodological aspects of our study. We have expanded our Methods section to provide more comprehensive information on both cancer status determination and sampling weight application: "Cancer status in our study was determined using the NHANES Medical Conditions Questionnaire (MCQ), which collected self-reported data on medical conditions through personal interviews conducted by trained NHANES staff. Specifically, participants were asked "Has a doctor or other health professional ever told you that you had cancer or a malignancy of any kind?" (variable MCQ220). Those responding "Yes" were classified as having a cancer history, while those responding "No" formed our non-cancer group. Participants with "Don't know" or refused responses were excluded from our analysis. For participants confirming a cancer history, additional information was collected regarding cancer type and age at diagnosis. We excluded individuals with lung cancer to avoid potential confounding with our primary outcome measure."

“To account for NHANES complex survey design and ensure nationally representative estimates, we applied appropriate sampling weights following NHANES analytical guidelines. Since our analysis combined data from six survey cycles (2007-2018), we created modified sampling weights by dividing the original 2-year examination weights (WTMEC2YR) by 6. For multivariable analyses, we incorporated the stratification variable (SDMVSTRA) and primary sampling unit (SDMVPSU) along with these modified weights using the svydesign function in R with the following specification: study_design <- svydesign(id = ~SDMVPSU, strata = ~SDMVSTRA, weights = ~WTMEC2YR, data = data, nest = TRUE). This approach accounts for the differential selection probabilities, non-response adjustments, and post-stratification alignments to census population totals while properly adjusting standard errors for the complex sampling design.”

4. Make minor grammatical and language edits to improve sentence clarity and overall flow. We have completed a thorough review of the manuscript and made all necessary grammatical and language edits to improve sentence clarity and overall flow. We carefully addressed each instance of awkward phrasing, refined transition sentences between sections, standardized terminology throughout the text, and ensured consistent verb tense usage. These revisions have enhanced the readability and precision of our work while maintaining the scientific integrity of our findings. We sincerely appreciate the reviewer's attention to these important details, as clear scientific communication is essential for effectively sharing our research contributions with the scientific community. We extend our heartfelt thanks for this valuable feedback, and if there are any remaining concerns about the language or presentation, we would be more than happy to address them in further revisions.

---

## [Editor Report · Decision Letter 2]

Cancer Disrupts Sex Hormone-Inflammation Relationships: Analysis of ALI in Males from NHANES 2007-2018

PONE-D-25-15723R2

Dear Dr. Yicong Zhou,

We’re pleased to inform you that your manuscript has been judged scientifically suitable for publication and will be formally accepted for publication once it meets all outstanding technical requirements.

Kind regards,

Akingbolabo Daniel Ogunlakin, Phd

Academic Editor

PLOS ONE
---

## [Editor Report · Acceptance letter]

PONE-D-25-15723R2

PLOS ONE

Dear Dr. Zhou,

I'm pleased to inform you that your manuscript has been deemed suitable for publication in PLOS ONE. Congratulations! Your manuscript is now being handed over to our production team.

Kind regards,

on behalf of

Dr. Akingbolabo Daniel Ogunlakin

Academic Editor

PLOS ONE